# Rare Earth Element Content in Hair Samples of Children Living in the Vicinity of the Kola Peninsula Mining Site and Nervous System Diseases

**DOI:** 10.3390/biology13080626

**Published:** 2024-08-17

**Authors:** Natalia K. Belisheva, Svetlana V. Drogobuzhskaya

**Affiliations:** 1Research Centre for Human Adaptation in the Arctic, Federal Research Centre “Kola Science Centre of the Russian Academy of Sciences” (RCHAA KSC RAS), Akademgorodok, 41a, 184209 Apatity, Russia; 2Tananaev Institute of Chemistry—Subdivision of the Federal Research Centre “Kola Science Centre of the Russian Academy of Sciences”, Akademgorodok, 26 a, 184209 Apatity, Russia; s.drogobuzhskaia@ksc.ru

**Keywords:** rare earth elements, nervous system, paroxysmal disorders, cerebral palsy, epilepsy, children

## Abstract

**Simple Summary:**

Rare earth elements (REEs) are widely used in various fields of human activity. However, data from many studies indicate the harmful effects of REEs on humans and animals, including the nervous system. In this study, we estimated the REE content in hair samples of children living on the Kola Peninsula (Murmansk region) in the vicinity of the Lovozersky REE Mining and Processing Plant, and we linked the REE content in children’s bodies to the prevalence of the nervous system diseases in Lovozersky District. We observed a higher prevalence of nervous system diseases (episodic paroxysmal disorders (G40–G47), cerebral palsy (G80–G83), epilepsy and status epilepticus (G40–G41)) in children aged 15–17 years in Lovozersky District compared to other territories of the Murmansk region. Indirect evidence of the link between REE content in hair samples of children and nervous system diseases is provided by data in the literature. According to these data, REEs are deposited in the brain structures responsible for the function of the nervous system, thus impairing them. This investigation highlights the relevance of studying the role of REEs in nervous system diseases, and it points to the need for the further research in this area.

**Abstract:**

The aim of this study is to assess the rare earth element (REE) content in hair samples of children living in Lovozero village, near an REE mining site, and the possible effects of REEs on the prevalence of nervous system diseases in Lovozersky District (Murmansk region, Kola Peninsula). Fifty-three school-age children were recruited for the analysis of REE content in hair samples. REE (Y, La, Ce, Pr, Nd, Sm, Eu, Gd, Tb, Dy, Ho, Er, Tm, Yb, and Lu) content was estimated by means of inductively coupled plasma mass spectrometry (ICP-MS). The analysis of REE content in the hair of children living in Russia, Kazakhstan, and China indicated REE intake from the environment. The possible contribution of REEs to nervous system disorders is supported by the link between the REE content in hair samples of children living near REE mining areas (China) and the manifestation of cognitive disorders in these children. It is also found that the prevalence of nervous system diseases in children aged 15–17 years is higher in Lovozersky District compared to the other districts of the Murmansk region. In this paper, the possible contribution of REEs to the prevalence of episodic paroxysmal disorders (G40–G47), cerebral palsy (G80–G83), and epilepsy and status epilepticus (G40–G41) is discussed.

## 1. Introduction

Currently, rare earth elements (REEs) are widely used in various fields of human activity, and the demand for them is constantly growing. REE mining is limited to certain countries, with China being the leader among them. Russia also makes a certain contribution to the mining of REEs worldwide, with the main REE ore mines located on the Kola Peninsula and in Zabaikalsky Krai. REEs are a series of seventeen metallic elements in the periodic table, including lanthanides (Ln), scandium (Sc), and yttrium (Y) [1,2,3]. REEs are usually divided into two groups based on their different structures and characteristics. Lanthanum (La), cerium (Ce), praseodymium (Pr), neodymium (Nd), promethium (Pm), samarium (Sm), and europium (Eu) are referred to as light rare earth elements (LREEs), while gadolinium (Gd), terbium (Tb), dysprosium (Dy), holmium (Ho), erbium (Er), thulium (Tm), ytterbium (Yb), lutetium (Lu), Sc, and Y are referred to as heavy rare earth elements (HREEs) [1,2,3]. 

REE mining is accompanied by the distribution of REEs in the air, soil, water, and agricultural products; the biological effectiveness of REEs and the assessment of their possible negative effects on the human body are thus gaining more attention [1,2,3,4]. The human body can be exposed to REEs through various routes, such as inhalation, ingestion, dermal contact, and iatrogenic exposure. The intake of REEs from the environment causes their deposition in the body, which destroys the structure and functions of various organs in humans and induces multi-system diseases [1,2,5,6,7,8,9,10,11,12]. 

The results of epidemiological investigations in regions with a high background content of REEs, such as southern China, imply that REEs might be neurotoxic [1,5]. Observational studies of vulnerable populations, such as children and pregnant women in mining areas, have shown that REEs can lead to reduced intelligence and motor ability in children, be deposited in fetal brains, and affect neural tube development [13,14,15,16,17,18,19,20]. These studies showed that REEs can impair the development of the central nervous system (CNS) and even pass through the placental barrier, generating transgenerational inheritance [1]. 

Estimations of REE content in the hair samples of children and teenagers in Madagascar also showed the importance of studying REE accumulation in the bodies of children living in areas that are not thought to have deposits of these elements [21]. It is possible that exposure to REE in such areas could cause no less neurotoxic effects than in areas, where REE are mined.

These studies and investigations on animals show that REEs can cause damage to the CNS, but their underlying mechanisms and targets are still unclear [5,6,13].

The study of the toxicological effects of REEs is most relevant in those regions in which REE ore is mined. In this respect, the Kola Peninsula is a natural laboratory that enables the study of REE’s effects on human health. The Kola Peninsula is the most industrial territory in the Arctic region. The high density of industry and cities in the central part of the peninsula creates ideal conditions for the widespread distribution of pollutants [22,23]. The Lovozersky Mining and Processing Plant, extracting REEs from ore, is located near the Revda settlement, where miners and their families live. The Lovozero village is located 18.5 km from the Revda settlement and has no industrial activity.

The impact of the Lovozersky Mining and Processing Plant on the environment has been assessed in separate studies [24,25,26,27,28,29], which found that dust tail the processing plant is a major source of the environmental contamination near the plant, including Lovozero village [29]. 

Working in recycling facilities and living in the vicinity of mining sites leads to a significant increase of the intake of REEs [2,29,30,31,32,33,34,35,36,37]. The human body is exposed to various chemicals simultaneously [38,39]; therefore, human health in Lovozersky District is affected by a combination of all of the REEs contained in the loparite ore. Despite the possible neurotoxic effects of REEs, REE content in the bodies of the residents of Lovozersky District has never been assessed. In addition, no attempt has been made to link REE contamination in the territory of Lovozersky District with the prevalence of the morbidity of nervous system diseases in the population. 

Since children’s bodies are the most sensitive to REE exposure, the aims of our study were to assess the REE content in hair samples of children living in Lovozero village and compare the incidence of nervous system diseases in Lovozersky District with that in other districts of the Murmansk region, as well as to theoretically demonstrate a possible link between the REE content in children’s bodies and nervous system diseases.

## 2. Experimental Procedure 

### 2.1. Sample Preparation

Children aged 7–15 years (53 persons) living in the vicinity (Lovozero village) of the Kola Peninsula mining site, where REE-bearing ore is mined and processed, were recruited in this study. Hair sampling and preliminary preparation of these samples for REE analysis were carried out according to the recommendations of the IAEA [39]. The inclusion criteria for the children was parental consent for the study, and all children were presumptively healthy. Collected hair samples were prepared according to the procedure described in [40], in which the samples were washed with acetone and deionized water (18.2 MΩ/cm^2^) and dried at an ambient temperature. Demineralized water was obtained using an Element system (Millipore, Burlington, MA, USA), and the acid was purified using a BSB-939-IR unit (Berghof, Eningen, Germany). Hair samples (0.05–0.15 g) were treated with 10 mL of 65% pure nitric acid in a closed 50 mL polypropylene volumetric tube in a water bath at a temperature of 100°C for 1–2 h. After cooling, the digested solution was diluted with 20–50 mL of 2% HNO_3_ in a volumetric flask. Working solutions were freshly prepared daily for analysis and then stored in 15 mL polypropylene centrifugal tubes for measurements (Figure 1).

### 2.2. Sample Analysis

All samples were analyzed for REEs (Y, La, Ce, Pr, Nd, Sm, Eu, Gd, Tb, Dy, Ho, Er, Tm, Yb, and Lu) and other elements in the Laboratory of Optical and Chemical Analysis methods of the Tananaev Institute of Chemistry Russian Academy of Sciences (ICT KSC RAS). Inductively coupled plasma mass spectrometry (ICP-MS) was used in this study. Elemental ICP-MS analysis (ELAN 9000 DRC-e, Perkin Elmer, Waltham, MA, USA) of samples was carried out using the available methods [41,42] and the methods developed in the ICT KSC RAS. 

The following conditions were maintained for all measurements: a plasma gas flow of 15.0 L min^−1^, a nebulizer gas flow of 0.90 L min^−1^, an auxiliary gas flow of 1.25 L min^−1^, and a radio frequency (RF) power of 1.3 kW. Multi-element calibration solutions from Inorganic Ventures (Christiansburg, VA, USA) (IV-STOCK-21, IV-STOCK-26, IV-STOCK-28, IV-STOCK-29) were used for the instrumental external calibration, and the linear fitting rates were 0.9999. Potential spectral interference was recognized during the validation of the method, and mathematical methods of interference correction were chosen. The limit of determination of elements was in the range of 0.001 to 0.0001 mg kg^−1^, depending on the determined element of study. The uncertainty level was estimated for the procedure, including sample preparation at the level of 10–15%. Quality control of the accuracy and precision of the analysis was carried out using the analysis of international and Russian reference materials certified for the content of the determined elements. 

### 2.3. Comparison Areas

The morbidity of children in Lovozersky District, where the dominant source of contamination is dust tail containing REEs, was compared with the corresponding indices at the sites given in Figure 2. 

1. Murmansk city, where there are no mining facilities, but there are a wide range of pollutants typical in urbanized areas. 2. Apatity city, which is in the sphere of dusting of the “ANOF-2” tailing dump—one of the largest sources of environmental pollution on the Kola Peninsula. Suspended substances include SiO_2_, TiO_2_, Al_2_O_3_, Fe_2_O_3_, FeO, P_2_O_5_, CaO, SrO, MgO, MnO, Na_2_O, K_2_O, and F_2_ [43]. 3. Kandalaksha city, where emissions from the Kandalaksha aluminum smelter contain a significant amount of such pollutants as hydrogen fluoride, fluorides, polycyclic aromatic hydrocarbons, and inorganic dust. Moreover, the most significant pollutants are compounds, the presence of which in the atmosphere can have a significant impact on human health and, in particular, the morbidity of children [22,23]. 4. Monchegorsk city, where the main pollutant is sulfur dioxide in emissions from the Kola Mining and Metallurgical Company, a division of Norilsk Nickel. Sulfur dioxide causes irritation of the skin and mucous membranes in the nose, eyes and upper respiratory tract. 5. Olenegorsk city, where a mining and processing plant is located: hematite–magnetite ores of the Olenegorsky deposit belong to ferruginous quartzites with a content of the following elements in magnetite: Fe, 72.1 ± 0.08%; Mn, 0.02–0.04%; and negligible contents of Al, Si, Cu, and Mg. 6. Kovdorsky District, with an ore mining and processing industrial complex that is Russia’s second-largest producer of apatite concentrate in terms of production output and the world’s only producer of baddeleyite concentrate (ZrO_2_). The iron mine processes magnetite-rich ore, and the mineral composition of tailings from the Kovdorsky GOK enrichment plant is as follows: CaO—27.1%; MgO—21.1%; SiO_2_—18.2%; P_2_O_5_—10.5%; Al_2_O_3_—2.2%; Fe_2_O_3_—2.0%; Na_2_O—0.8%; K_2_O—0.8%; SO_3_—0.5%; MnO—0.1%; TiO_2_—0.1%; others—16.6%. 7. Kolsky District, characterized by mixed types of pollution caused by the transportation of air masses containing toxic compounds from the main sources of contamination on the Kola Peninsula. 8. Pechenganickel Combine, where the dominant nickel concentrator in the ore is pentlandite. Its chemical composition in ore rocks, on average, is as follows: Ni—35.3%; Fe—30.8%; Co—0.7%; S—33.2%; sum—99.9%; formula: (Ni_4.65_Fe_4.26_Co_0.08_)_8.99_S_8.00_. In its disseminated form, its chemical composition is as follows: Ni—34.1%; Fe—32.0%; Co—0.6%; S—33.0%; sum—99.7%; formula: (Ni_4.50_Fe_4.44_Co_0.08_)_8.02_S_7.98_.

Thus, similarities and differences in the morbidity of the child population living in these territories will make it possible to identify the common and specific possible causes of morbidity in children due to the nature of local contamination in the compared areas.

### 2.4. Analysis of the Population Morbidity in the Kola North

The Statistical Compendiums “Incidence of the population of the Murmansk region 1995–2018” as well as data from the Murmansk Regional Medical Information and Analytical Center for 2018–2020 were used in this work. 

### 2.5. Statistical Data Processing 

The data obtained were statistically processed by using the STATISTICA 10 software package; graphing was carried out using the graphic editor ORIGIN. The normality of the distribution of each element was determined using the Shapiro–Wilk method. Statistical characteristics included the average arithmetic values of the studied parameters (*M*), the statistical error (*±SE*), and the standard deviation (*±SD*). Along with these, medians (*Me*), the variation spread (*Min*–*Max*), percentile (*P25*–*P75*) and quartile (*Q25*–*Q75*) ranges, and coefficients of variation (*C.V.*) were also calculated. 

## 3. Results

### 3.1. REE Content in Children’s Hair Samples

The statistical characteristics of the content of REEs in the hair samples of children living in Lovozero village in the vicinity of the mining facility are presented in Table 1.

In Table 1, one can see that Sc, Dy, and Yb have the highest coefficients of variability for their content in hair samples of children. The uneven accumulation of these elements in different children may be associated with differing exposure to these REEs as a result of unknown reasons. Since the REE content in hair samples from children depends on the area of residence, the proximity to ore deposits containing REEs, their output into the environment, and their concentrations in the soil, water, air, and foodstuffs [1,2,3], there are no standart reference data for REE content in hair samples of children. The comparison of REE content in hair samples of children from Lovozero with those in hair samples of children living in other territories could prove that REE content is increased by nearby mining facilities. Accordingly, we hypothesize that REE accumulation with age may be associated with an increase in the prevalence of nervous system diseases.

### 3.2. Comparison

Table 2 shows the contents of some trace elements and REEs in the hair samples of children from Lovozero village, Zabaikalsky Krai [44], and the Pavlodar region in the Republic of Kazakhstan [45], as well as in hair samples of adults from Sweden [46].

The territory of Zabaikalsky Krai is characterized by deposits of silver–lead–zinc and polymetallic gold, tungsten, tin, molybdenum, uranium–gold, and lead–zinc ores, which is similar to the Kola Peninsula in terms of the diversity of deposits [43]. The Pavlodar region in Kazakhstan was in the zone of influence of the former nuclear test site at Semipalatinsk (https://infourok.ru/material.html?mid=53145, accessed on 25 July 2024). Kazakhstan also has 56 explored uranium deposits, 14 of which are currently being mined (https://orda.kz/kazahstanskij-uran-milliardy-v-zabroshennye-shahty-i-nesankcionirovannye-perevozki/, accessed on 25 July 2024). Thus, a comparison of the content of elements in hair samples of children in territories with that in contrasting geochemical structures allows us to discover the peculiarities of element accumulation depending on their presence in the environment. Unfortunately, we were unable to find studies on REE content in hair samples from children in background areas in the Euro-Arctic region. Therefore, the REE content in the hair samples of Swedish residents was taken as a conditional background value.

In Table 2, the mean with the standard deviation (*M ± SD*), the median with the inter quartile range (*Me* (*Q25*–*Q75*)), the mean with the standard error (*M ± SE*), and the mean with the standard deviation and with the median (*M* (*SD*) *ME*) are shown in order to compare the elemental content in hair samples of children from Lovozero with the corresponding data in the cited literature [43,44,45]. 

The data presented in Table 2 were obtained by means of two methods: those for Lovozero (1) and Sweden (4) were obtained by using ICP-MS, while those for the Zabaikalsky region (2) [44] and Kazakhstan (Pavlodar District) [45] were obtained by using multi-element instrumental neutron activation analysis (INAA). The comparability of these methods can be seen, for example, by comparing the chromium (Cr) content in the hair samples from Lovozero and from Zabaikalsky Krai: 4.43 (1.32–6.5) and 4.68 (0.46–6.11) µg/g, respectively. These contents are 8.9- and 9.4-fold higher than those in the hair samples of children from Kazakhstan, and 26.5- and 28.0-fold higher than those in the hair samples of Swedish residents, respectively (Table 2). The deposits of chromium ores in the Karelian-Kola Province and in the Zabaikalian Province might be the cause of the increased Cr content in the hair samples of children from Lovozero and Zabaikalsky Krai. Therefore, the data in Table 2 indirectly indicate the comparability of these two methods of analysis, adequately detecting the presence of Cr in the environment. 

The accumulation of chemical elements in the hair samples of children from Zabaikalsky Krai and the specificity of their distribution depend on natural and technological factors [44]. For example, the content of iron (Fe) exceeds the corresponding content in hair samples of children from Lovozero 34.5-fold, in those from Kazakhstan 17.6-fold, and in hair samples of Swedish residents 82.6-fold. The high Fe content in the hair samples of children from Zabaikalsky Krai probably relates to the substantial iron ore deposits in this territory (https://tass.ru/ekonomika/20438599, accessed on 25 July 2024).

The content of silver (Ag) in the hair samples of children from Zabaikalsky Krai is 8.4 times higher than that in the hair of children from Lovozero, 7.8 times higher than that in the hair of children from Kazakhstan, and by 10.2 times higher than that in the hair of Swedish residents. The content of gold (Au) in the hair samples from Zabaikalsky Krai is 2.5 times higher than that in the hair samples from Lovozero, but its content is 3.7 times lower than that in the hair samples from Kazakhstan and is comparable with the content in the hair samples of Swedish residents. The higher content of Au in the hair samples of children from the Pavlodar region (Kazakhstan) is most likely due to the presence of gold deposits in this area (https://webmineral.ru/deposits/item.php?id=2858, accessed on 25 July 2024). These findings further emphasize the exceptional role of elemental content in children’s hair as an indicator of their presence in the environment.

Significant variations can be observed when comparing the lanthanide (Ln) content in the hair samples of children from Lovozero with the hair samples of residents of the comparison areas. However, the content of lanthanum (La) is comparable in the hair samples of residents of all areas. The median values of europium (Eu) in the hair samples of children from Lovozero and from Zabaikalsky Krai are comparable, but lower than those in the hair samples of children from the Pavlodar region. The content of neodymium (Nd) is lower in the hair samples of children from Lovozero than in the hair samples of children from Zabaikalsky Krai. However, the content of samarium (Sm) is almost four times higher than that in the hair samples of children from Zabaikalsky Krai, and it is 1.6 times lower than that in children from the Pavlodar region. The content of ytterbium (Yb) in the hair samples of children from Lovozero is 1.7 times higher than that in the hair samples of children from Zabaikalsky Krai, while it is 2.5 times lower than that in the hair samples of children from the Pavlodar region. The content of lutetium (Lu) is comparable among the hair samples of children living in the comparison areas. The contents of hafnium (Hf) and thallium (Ta) are lower in the hair samples of children from Lovozero than in the hair samples of children from the Zabaikalsky Krai and Pavlodar regions, and they are comparable with the content in the hair samples of Swedish residents.

A comparison of the contents of other elements is not provided in this text; we have noted the most important elements characterizing the specificity of the comparison areas. For example, the content of uranium (U) in the hair samples of children from the Pavlodar region is 66 times, 33 times, and 9.2 times higher than that in children from Lovozero and Zabaikalsky Krai and Swedish residents, respectively, indicating the area’s possible history of contamination with natural and technogenic radionuclides. 

Thus, we compared the content of elements in hair samples of children living in areas with contrast distributions of elements in the environment. Moreover, we clearly showed the correspondence between the accumulation of elements in children’s bodies and the content of these elements in the environment. We also showed that the content of some REEs may be higher in residents of areas where there are no REE mines. In this respect, our data are in good agreement with studies carried out in Madagascar, which show the presence of REEs in the hair of children living in areas where REE mining is absent [21]. However, the bio-effects of REEs should be more significant in areas where REEs are mined, since their effects on the nervous system are more pronounced with an increase in exposure to REEs [14,20].

Chinese researchers have extensive experience in studying the relationship between REE contamination and the cognitive function of the population living in REE-contaminated areas [15,16,17,18,19,34]. Extrapolating data on the neurotoxicity of REEs, found in [5,6,13,14,20,34], to assess the possible effects of REEs associated with nervous system diseases in children, we compared the REE content in hair samples of children from Lovozero and children from China, living at different distances from REE mines [34] (Table 3). We also compared the REE content in hair samples of children in areas where children have manifested cognitive impairments with the REE content in hair samples of children from Lovozero (Table 4) [17].

Comparisons of REE content in the hair samples of children from Lovozero village (L) with the hair samples of children from different villages in China (M1, M2, C1, and C2) are shown in Table 3. The peculiarities of the territories in China were associated with the mining of ore containing LREEs. In a LREE mining area in southern China, Shang village (referred to as M1) and Liao village (referred to as M2) were selected as sampling sites [33]. There were many open-casting mining surfaces around and near M1, and M2 was relatively farther away from a few mining surfaces. Two other villages, Dong village (referred to as C1) and Ping village (referred to as C2) in an area with no mining activity in the same county, were selected as control villages. C1 village was nearer to the mining area (with a direct distance of 8 km) than C2, which was 14 km from the mining area and separated by high mountains. A total of 323 healthy students aged 11–15 years at the primary schools in the above-mentioned villages were chosen as the subjects of this study [34]. 

It can be seen that the content of all light elements in hair samples of children from Lovozero is lower than in the hair samples of children from China (Table 3, Figure 2).

The diagram in Figure 3 shows the distribution of REEs (geometric mean of the samples) in the hair samples of children living in Lovozero village and in the hair samples of children from different villages in China. The hair samples of children living in China are rich in LREEs, while HREEs predominate in the hair samples of children from Lovozero. At the same time, the contents of HREEs (from gadolinium (Gd) to Yb) in the hair samples of children from Lovozero are comparable with the corresponding contents in the hair samples of children from Liao (M2), Dong (C1), and Ping (C2) villages. Moreover, the content of Lu in the hair samples of children from Lovozero exceeds the corresponding values in the hair samples of children from the comparison areas (M1, M2, C1, and C2). In addition, it can be seen that the content of almost all LREEs is higher in Shang village (M1) (Figure 3), where many open-casting mining surfaces around and near the site are present [34]. The presented results once again show that living in the vicinity of ore mining sites containing REEs leads to REE accumulation in the bodies of the residents of the neighboring territories and, in particular, in children’s bodies. Regarding this connection, it can be seen that the REE content in the hair samples of children from Lovozero corresponds to the elemental composition of the ore mined and processed by the Lovozersky Mining and Processing Plant. In addition, the predominance of HREEs in the hair samples from children in Lovozero in comparison with those from China demonstrates the difference between the REE content of the ore mined in the Lovozersky region and that mined in the corresponding territories in China.

Further, we compared the REE content in hair samples of children from Lovozero village and children from another REE mining area, namely the Maoniuping rare earth deposit, which is located in Mianning City, Sichuan Province, China [17]. Four villages around this REE mining area were selected in this research. Exposure to REEs and lead was assessed by analyzing their content in the scalp hair samples of 95 children aged 6–16 years who lived in the REE mining area. This study showed that both light and heavy REEs, as well as lead (Pb), detected in the hair samples of the children, were associated with a negative effect on their intelligence [17].

A comparison of the REE content in children’s hair samples from Lovozero village with the content of corresponding elements in the hair samples of children from the mining area in Mianning City is presented in Table 4.

The content of LREEs in the hair samples of children from the mining area in Mianning City was higher compared with the REE content in the hair samples of children from Lovozero, as shown in Table 4. The content of Eu in the hair samples of children from Lovozero is slightly lower than in the hair samples of children from Mianning City. However, the content of HREEs is higher in the hair samples of children from Lovozero than in the corresponding hair samples of children from Mianning City. Thus, the mean contents of Tb, Ho, Er, Tm, Yb, and Lu in the hair samples of children from Lovozero exceed the corresponding values in children from the mining area of Mianning City by 1.6, 1.65, 1.5, 3.0, 1.9, and 4.6 times, respectively, and the contents of Tb, Dy, Ho, Er, Tm, Yb, and Lu are higher in 25 percent of children from Lovozero (P25) than in children from this mining area in China by 2.8, 1.2, 1.9, 2.4, 4.6, 2.4, and 7.3 times, respectively. 

Since HREEs, independently of LREEs and lead, contributed to the decrease in intelligence of children from this mining area in China [17], we assume that the REE content in the bodies of children from Lovozero might also affect on their cognitive function and other functions of the nervous system.

### 3.3. Nervous System Diseases

The comparison of the prevalence of nervous system diseases in children aged 0–14 years with the corresponding prevalence of diseases in adolescents (15–17 years old) in the same territory could be indirectly indicative of the accumulation of REEs with age and the contribution of REEs to these diseases, if the morbidity of adolescents increases.

The average annual values (*M ± SE*) of the prevalence of nervous system diseases in children in the compared territories for the period of 2018–2020 are shown in Figure 4.

It can be seen that the prevalence of episodic paroxysmal disorders (G40–G47) is the highest in Murmansk city (28.5 ± 2.9), followed by the Lovozersky (16.9 ± 6.2) and Kolsky (14.0 ± 0.8) Districts, in children aged 0–14 years (Figure 4(A1)). However, the prevalence of these diseases is the highest in Lovozersky District (51.7 ± 4.4), followed by Murmansk city (42.6 ± 2.6) and Kolsky District (34.0 ± 3.2), among adolescents aged 15–17 years (Figure 4(A2)). Thus, the prevalence of episodic and paroxysmal disorders in adolescents aged 15–17 years is increased by 1.5, 3.1, and 2.4 times compared with children aged 0–14 years in Murmansk city, Lovozersky District and Kolsky District, respectively. 

The prevalence of cerebral palsy and other paralytic syndromes (G80–G83) in children aged 0–14 years old is highest in Kandalaksha District (5.3 ± 0.2), then in Murmansk (4.4 ± 0.2), Kolsky District (3.7 ± 0.1), and Lovozersky District (3.2 ± 1.0). However, the prevalence of these diseases among adolescents is the highest in Lovozersky District (6.0 ± 1.7), followed by Murmansk (4.7 ± 0.4), Kandalaksha District (4.6 ± 0.1), and Kolsky District (3.0 ± 0.8) (Figure 4(B1,B2)). These data demonstrate that the prevalence of cerebral palsy and other paralytic syndromes is 1.1 and 1.9 times higher in Murmansk and in Lovozersky District, respectively, and 1.2 times lower in the Kandalaksha and Kolsky Districts in adolescents compared to children aged 0–14 years in the corresponding areas. 

The prevalence of epilepsy and status epilepticus (G40–G41) in children aged 0–14 years is highest in Murmansk (7.3 ± 0.8), followed by the Kandalaksha (6.2 ± 0.3), Kovdorsky (5.6 ± 0.9), and Lovozersky (5.9 ± 1.2) Districts (Figure 4(C1)). However, the prevalence of these diseases among adolescents is highest in Lovozersky District (15.0 ± 3.3), followed by Kandalaksha (14.1 ± 3.0), Murmansk (9.0 ± 0.9) and Kovdorsky District (6.5 ± 1.0) (Figure 4(C2)). These data demonstrate the increase in the prevalence of epilepsy and status epilepticus among adolescents relative to children aged 0–14 years in Lovozersky District, by 2.5 times. At the same time, there is a 1.2-, 2.3-, and 1.2-fold increase in the prevalence of these disorders among adolescents relative to children aged 0–14 years in Murmansk, Kandalaksha, and Kovdorsky, respectively. 

Thus, Lovozersky District is a leader in the prevalence of nervous system diseases among adolescents aged 15–17 years compared with the other territories. Therefore, we suggest that the increase in the prevalence of nervous system diseases in children from Lovozersky District with age is caused by the chronic accumulation of lanthanides in low doses in children’s bodies, beginning in the antenatal period. REEs could be ingested by children and their mothers during pregnancy through water [24,29], via air particle inhalation [4,26,27], and even dermal absorption [7], as well as other routes.

## 4. Discussion

The comparison of the REE content in the hair samples of children from Lovozero village with the corresponding REE values in the hair samples of children living in other territories (Table 2, Table 3 and Table 4) shows the predominance of HREEs in the hair samples from Lovozero. To some extent, the comparability of the physiological effects of exposure to heavy and light REEs is similar to that established in the work of Wang et al. [17]. The similarity of the effects of HREEs and LREEs is also confirmed by the fact that Er^3+^, as a heavy REE, like La^3+^ [47,48,49,50,51,52,53,54,55,56,57,58] and Pr^3+^ [49], has effects on the conductivity of nervous impulses in the frog neuromuscular synapse [50]. In addition, Haley et al. [50,51,52,53,54] investigated the pharmacological and toxicological effects of light and heavy REEs, including Sc, Pr, Nd, Sm, Gd, Tb, Dy, Ho, Er, Tm, and Yb, on animals. All REEs had a similar acute toxicity and similar pharmacological effects; moreover, the acute lethality increased with the atomic number of the REE [51]. Hence, the biological effects associated with exposure to La and other LREEs could be induced by exposure to HREEs. Is there any reason to assume that REE accumulation in children might lead to the high prevalence of nervous system diseases in Lovozersky District?

### 4.1. Accumulation of REEs in the Brain

REEs cross the blood–brain barrier and accumulate in the brain tissue [1,2,5,9,11,19,45,56,57,58,59]. REEs can also penetrate the placental barrier and enter into the fetal body and brain tissue, and young animals can ingest REEs via breastfeeding [14,20,57]. Moreover, REEs can be transported into the brain through cerebrospinal fluid (CSF) [59]. 

Numerous studies have demonstrated the effects of REEs on the nervous system. Zhu et al. [19] found the cumulative effect of REEs on brain function. The daily supplementation of REEs in rat diets decreased acetylcholinesterase (AchE) activity in the hippocampus compared with the control group. Swelling in the cell bodies of the cerebral neurons and the phagocytic phenomenon were observed in rats receiving a moderate dose of REEs [19]. The consequences of REE accumulation in the cerebral cortex, which occurs as a result of long-term intake from the environment in small amounts, could cause subclinical damage [19]. A study by Feng et al. [6] revealed that La^3+^accumulated in the brain and remained for a long time after rats were exposed to LaCl_3_ through oral administration. Significantly higher concentrations of La accumulated in the cerebral cortex, hippocampus, and cerebellum [6]. 

The high affinity of Ln for calcium ions leads to Ca^2+^ substitution in the Ca^2+^-binding sites of proteins and to changes in neuronal conductivity. Changes in Ca^2+^ metabolism under the influence of Ln can have serious consequences for brain function [60]. La^3+^ can inhibit calcium-dependent neurotransmitter release [61], bind with the Ca^2+^ binding sites on the axon membrane, and alter membrane conductivity [62]. La^3+^ reduces both the influx and efflux of Ca^2+^ in the squid giant axon [63] and inhibits Ca^2+^ binding with brain synaptosomes simultaneously with the suppression of the activities of neural ATPase and AchE [64]. The rate of high-affinity uptake of ^14^C glutamate by brain synaptosomes depletes after acute LaCl_3_ toxicity in chicks [65]. La^3+^ also prevents the electrically stimulated release of transmitters from synapse of the squid giant axon [66] and the release of neurotransmitter amino acids from the spinal–medullary synaptosomes of rats [67]. La^3+^ causes a large increase in miniature end plate potential discharge from the frog neuromuscular junction [47]. La^3+^, Er^3+^ and Pr^3+^ also exhibit dual action on transmitter release at the frog neuromuscular synapse [68]. 

Significant increases in the La content in the serum, in the hippocampus, and in the cerebral cortex; a rise in intracellular free calcium in hippocampal cells; a significant decrease in the activity of Ca2þ-ATPases; and the inhibition of protein kinase-1 activities and the signaling pathways in the hippocampus were observed in rats by He et al. after long-term LaCl_3_ exposure [5]. Yang et al. [69] found that LaCl_3_ negatively affects the synaptic ultrastructure in the hippocampus. Heuser J.E. [70] also revealed an altered synaptic ultrastructure in hippocampal neuronal cell cultures after exposure to LaCl_3_. A study of the effects of LaCl_3_, CeCl_3_, and NdCl_3_ on mouse brains showed that injury to the brain and oxidative stress caused by Ln trigger a cascade of reactions, such as lipid peroxidation, decreases in the total antioxidative capacity and activities of antioxidative enzymes, the excessive release of nitric oxide, an increase in glutamic acid contents, and downregulated levels of acetylcholinesterase activities [71]. The accumulation of Ce in the mouse hippocampus causes hippocampal apoptosis and impairs the spatial recognition memory of mice [72].

Heavy REEs exhibit a distribution that is similar to that of La in the brain [55]. Intravenous injections of ^169^Yb in rats allowed for an assessment of its distribution in five brain regions (hypothalamus, cerebellum, hippocampus, corpus striatum and cerebral cortex). It was found that Yb did enter the brain, where it remained for a long time. The highest specific activities were observed in the hypothalamus, hippocampus and cerebellum. The ^153^Sm distribution in the brain was similar to that of ^169^Yb [73]. However, unlike ^169^YbCl_3_, the highest amounts and the lowest concentrations of ^153^Sm were found in the cerebral cortex and the highest concentrations of ^153^Sm were found in the hypothalamus (Notation: the amount is the absolute content of ^169^Yb in the cerebral cortex, and concentration is the relative content - quantity (amount) per tissue volume)

Gadolinium (Gd) also accumulates in the brain, along with other HREEs. Usually, gadolinium-based contrast agents (GBCAs) are used in MRI [9]. Autopsy specimens have shown that GBCAs accumulate in the dentate nucleus, globus pallidus, cerebellar white matter, frontal lobe white matter, and frontal lobe cortical matter. A higher Gd signal intensity was detected in the dentate nucleus and globus pallidus [74,75,76,77]. 

In addition to soluble REE forms, micro-sized Ln particles can also penetrate the brain. Intravenous injections of Sc_2_O_3_ with a particle diameter of ~3 µm in rats revealed the distribution of Sc particles in the brain [78]. While Yokel et al. [79] highlighted the minor penetration of Ce nanoparticles into the brain parenchyma, Baranovskaya et al. [56] found good evidence of Ce’s penetration into the brain based on the presence of cerium-containing mineral phases in the brain of Far East deer using electron microscopy.

Along with the brain neurons, disorders in the neuroglia (astrocytes) due to Ln exposure are linked to the occurrence of nervous system diseases [80]. LaCl_3_ can also induce axon abnormality in the hippocampus and primary cultured neurons [81].

The effects of REEs on the nervous system can also occur through changes in gene expression. In particular, Y^3+^can change the expression of some genes, which may be responsible for the toxic effects of REEs on learning and memory [82].

Thus, there is no doubt that REEs, upon introduction into an organism, can penetrate the brain in various ways and accumulate in different brain structures responsible for thinking, long-term and short-term memory, sensory perception, and motor activity. The disturbance of these structures as a result of REEs could lead to such diseases as paroxysmal activity, paralysis, and epilepsy.

### 4.2. Nervous System Diseases and REEs

The increasing use of rare earth elements (REEs) has resulted in a considerable risk of environmental exposure [1,2,3]. At the same time, the adverse effects of REEs on reproductive health [83], including influence on prenatal and on children’s neurodevelopment, are not yet fully recognized [14,20,57]. However, there is clear evidence that REEs, among other factors, could contribute to diseases of the nervous system.

#### 4.2.1. Episodic Paroxysmal Disorders (G40–G47)

Paroxysmal movement disorders (PMDs) are neurological diseases typically manifesting in intermittent attacks of abnormal involuntary movements [84,85]. There are two main recognized categories of PMDs based on the phenomenology: paroxysmal dyskinesias (PxDs), characterized by transient episodes of hyperkinetic movement disorders, and episodic ataxias (EAs), which are characterized by attacks of cerebellar dysfunction [86,87]. PxDs are network disorders; either primary striatal dysfunction or an aberrant cerebellar output conveyed to the striatum results in this phenotype. The striatum (a cluster of neurons in the subcortical basal ganglia) is a critical component of the motor and reward systems; it receives glutamatergic and dopaminergic inputs from different sources and serves as the primary input to the rest of the basal ganglia. The disruption of synaptic neurotransmission, mostly affecting cerebellar and striatal circuitries, is a common final effect [87]. The exact pathophysiological mechanisms underlying PMDs are not fully understood. Studies have suggested that alterations in synaptic neurotransmission, brain energy metabolism, and ion channel function may all be involved in the pathogenesis of PMDs [86,88,89]. 

REEs may contribute to certain biological processes, leading to various hyperkinetic disorders and alterations. As shown above, REEs mainly accumulate in the cerebral cortex, hippocampus, and cerebellum. The main groups of neurons in which REEs have been detected include the dentate nucleus and globus pallidus. The dentate nucleus is a cluster of neurons in the cerebellum and it is the largest single structure linking the cerebellum to the rest of the brain [90]. The dorsal region of the dentate nucleus contains output channels involved in motor function [91,92]. The accumulation of REEs in the globus pallidus (GP), as a major component of the basal ganglia with principal inputs from the striatum and principal direct outputs to the thalamus and to the substantia nigra, may induce multiple lesions. Figure 5 shows the location of brain regions responsible for motor disorders and the deposition of certain REEs in these structures. 

#### 4.2.2. Cerebral Palsy and Other Paralytic Syndromes (G80–G83)

Cerebral palsy (CP) is primarily a neuromotor disorder that affects the development of movement, muscle tone, and posture [93,94,95]. The underlying pathophysiology is an injury to the developing brain in the prenatal through to the neonatal period [93]. The motor disorders of CP are often accompanied by disturbances of sensation, perception, cognition, communication, and behavior; by epilepsy; and by secondary musculoskeletal problems [93,94,95,96]. CP can be defined according to the anatomical site of the brain lesion (cerebral cortex, pyramidal tract, extrapyramidal system, or cerebellum) and clinical symptoms and signs (spasticity, dyskinesia (dystonic and choreo-athetotic forms), or ataxia) [95,96,97] (Figure 5).

In the majority of cases, there is evidence that factors present during the prenatal period play a prominent role, although the causes of congenital CP are unknown. CP is a motor impairment syndrome that results from a lesion occurring in the developing brain [95]. Prerequisites for the development of CP could lie early in the formation of the neural tube [98,99]. Most cases of CP result from an interference in brain development in utero. Ion channels participate in the formation of the brain and spinal cord during one of the first developmental steps, known as neural tube formation [99]. The excitable nature of neurons and muscle cells is dependent on the specific expression of ion channels and their subcellular localization in these cells [100]. During neural tube formation, neural plate cells exhibit Ca^2+^ transients partly mediated by the NMDA receptor, which is a glutamate receptor and predominantly a Ca^2+^ ion channel found in neurons [99]. The propagation of activity in neural tissue is generally associated with synaptic transmission, but the epileptiform activity of CP in the hippocampus can propagate with or without synaptic transmission due to the endogenous field effect transmission in the hippocampus [101]. The third to fourth week of embryogenesis is a critical period in the development of neural tube morphology. If any interference is encountered during this period, the neural tube may fail to close [98,102]. 

REEs can enter a fetus through the placental barrier [14,20,103], and exposure to REEs may affect the growth and development of the fetus. Cao et al. [20] studied the association between prenatal exposure to 13 REEs and the neurodevelopment of children at 24 months of age. It was observed that exposure to seven REEs during the first trimester was significantly associated with a decrease in the mental index (MDI), while exposure to nine REEs during the third trimester was significantly associated with decreased psychomotor development (PDI) scores in children at 24 months. Tm and Er in the first trimester and Ce and Ln in the third trimester displayed the greatest significance with regard to their joint effects on MDI and PDI, respectively. REEs may be a factor promoting impaired neural tube morphogenesis and the development of CP in children, since the third to fourth week (first trimester) is a critical period in the development of the neural tube. 

Wei et al. [14] examined the associations between the concentrations of 10 REEs in the maternal serum and the risk of fetal neural tube defects (NTDs). A higher level of La in the maternal serum was associated with an increased risk of fetal NTDs. When each of the 10 REEs were examined individually and when all 9 other REEs were simultaneously considered, the NTD risk increased with the overall concentrations of the 10 REEs under consideration as a co-exposure mixture [14].

The contribution of REEs to prenatal disorders leading to CP might manifest as alterations in ion channels [104,105] via abolishment of the “calcium response” of nerve terminals [68]. Imbalance in Ca and Ca^2+^ binding protein levels caused by La accumulation in the brain can disturb ion homeostasis and cause a series of physiological disorders in the CNC [47].

Some cases of CP that have a fetal origin are associated with non-cerebral congenital anomalies (CAs) [106,107,108,109,110,111]. Pharoah [106] found an elevated risk of CAs of the eye, cardiac anomalies, a cleft lip and/or palate, the congenital dislocation of the hips and talipes, and atresias of the esophagus and intestines. REEs appear to be able to contribute to prenatal development and promote CAs, associated with CP. ^141^Ce residues were distributed throughout the viscera and tissue, but their levels were higher in the eye, bone, testis, brain, heart, and adipose tissue, and cerium accumulation increased with an increasing dose and feed duration [112]. The exposure of pregnant women to Ce and Yb decreased thyroid-stimulating hormone (TSH) levels in infants [113], demonstrating the prenatal effects of Ce and Yb. The association between maternal exposure to REEs in the prenatal period and the prevalence of orofacial clefts (OFCs), as disruptions of the normal craniofacial structure, supports the possible involvement of REEs in the initiation of CAs [114].

La exposure influences learning and memory as well as the expression of apoptosis-related proteins in offspring rats [115]. La is transmitted to offspring rats through parental blood circulation and breast milk before delactation and through water drinking after delactation, and affects learning and memory and the hippocampal neurons of offspring rats [115]. LaCl_3_ impacts the dendritic spines in CA1 pyramidal cells, decreasing the dendritic spine density during development [116,117,118]. More than 90% of the excitatory synapses in the central nervous system are located on dendritic spines [119,120], and therefore, the effects of REEs in the neonatal period could induce lesions in neuron conductivity and promote the realization of PC by decreasing the dendritic spine density during development.

#### 4.2.3. Epilepsy and Status Epilepticus (G40–G41)

Epilepsy comprises a heterogeneous group of brain diseases which all share an enduring predisposition toward generating seizures [121]. The known causes of epilepsy include genetic abnormalities such as de novo mutations and/or precipitating injuries. In the majority of cases, the underlying causes remain elusive [122,123]. Epileptogenesis, a pathological process transforming a normal healthy brain into an epileptic brain, is characterized by multiple pathological changes within the brain, such as acute and ongoing cell death, aberrant synaptic reorganization and neurogenesis, blood–brain barrier (BBB) disruption, and inflammation, among many others [124]. Seizures are essentially a malfunctioning of the brain due to the “misfiring” of its neurons [124]. It is clear that neurons communicate through action potentials along their axons, and that these action potentials are electrical events that depend on the movements of ions, particularly sodium and potassium, across the neuronal cell membrane [124]. Zhang et al. [103] suggested a new propagation mechanism for neural activity in the hippocampus involving endogenous field effect transmission in epilepsy. The discovery that astrocytes release glutamate, which causes synchronous neuronal depolarizations, has led to the idea that a source of excitation in epileptic discharges could be these cells [125,126,127].

REEs disrupt astrocyte functions that may play a certain role in the provocation of epileptic seizures. Along with the brain neurons, disorders in the neuroglia (astrocytes) due to Ln exposure are of particular importance for the occurrence of nervous system diseases [80,81]. Astrocytes, according to current thinking, play a decisive role in nervous system diseases [125,126,127]. Because many of the synapses in the CNS are tripartite in nature, the disruption of astrocytic supportive functions and/or of gliotransmission has the potential to disrupt synaptic transmission, synaptic plasticity, and neuronal excitability [127,128]. The terminal foot of astrocytes is one important component of the blood–brain barrier, and La from the blood first encounters astrocytes [129]. La suppresses the astrocyte–neuron lactate shuttle (ANLS) in the rat hippocampus, and these changes in the ANLS system are probably linked to the neurotoxicity of La [80]. Astrocytes are the main target in toxic encephalopathies caused by environmental exposure to heavy metals [130]. 

## 5. Advantages of This Study

For the first time, the REE content in hair samples of children living in the vicinity of an facility that mines and processes REE-containing ore on the Kola Peninsula was assessed—in fact, this represents the first study of its kind in the Euro-Arctic region. Our data can serve as a reference for future studies comparing REE content in background and industrial areas. 

The association of qualitative and quantitative REE content in the hair samples of children with the incidence of nervous diseases in their area of residence poses serious challenges for biologists, physicians, and epidemiologists. Our study opens avenues for further studies into the mechanisms of REEs’ influence on the nervous system and emphasizes the need for more detailed studies on the symptomatology of nervous disorders in miners extracting REE-containing ore, as well as in residents of areas in the vicinity of REE mining facility. 

Our study may encourage epidemiologists to compare the morbidity of residents living in REE-contaminated areas in different regions and countries with that of residents of non-contaminated areas. Such studies will allow us to assess the real risk to the population posed by REE contamination and to develop measures to reduce the risk of morbidity from diseases associated with REE accumulation.

## 6. Conclusions

The REE content in hair samples of children from Lovozero village, which were similar to and exceeded the REE content in miners and children living in ore-mining zones in other territories, indicate that REEs are introduced into these children’s bodies from external sources. Theoretically, it is possible that the accumulation of REEs in children might occur in utero in cases where the mother resided in an area contaminated with REEs for a prolonged time. In addition, the gradual deposition of REEs in the children born and living in the contaminated territories could lead to an increase in the incidence of nervous system diseases with age due to the increased REE content in their bodies. The accumulation of REEs in certain brain structures could lead to impaired synaptic transmission, altered neurotransmission and cellular neuronal metabolism, the disruption of ion channels, and other phenomena underlying nervous system diseases. This study suggests the possible role of REEs in nervous system dysfunction, but obtaining proof of such a role requires further research. 

These studies should be comprehensive and interdisciplinary, involving in-depth medical neurophysiological examinations of the population living in the territories contaminated with REEs; the collection and analyses of samples of water, soil, and vegetation at different distances from the source of REE contamination; the formation of comparison groups of the population with different doses of REE exposure; the collection of hair samples and biological substrates from pregnant women, children, and miners; the detection of REE content in the collected material; analyses of the obtained materials; and the study of the contribution of REE content in the body to nervous system dysfunctions in the examinees.

## 7. Limitations

This study of REE content was limited to hair samples from children living in Lovozero village in the vicinity of a mining plant processing REE-containing ore. In the future, this study will be continued and expanded to include analyses of REE content in hair samples of women and men living in Lovozero village, as well as miners extracting ore.

## Figures and Tables

**Figure 1 biology-13-00626-f001:**
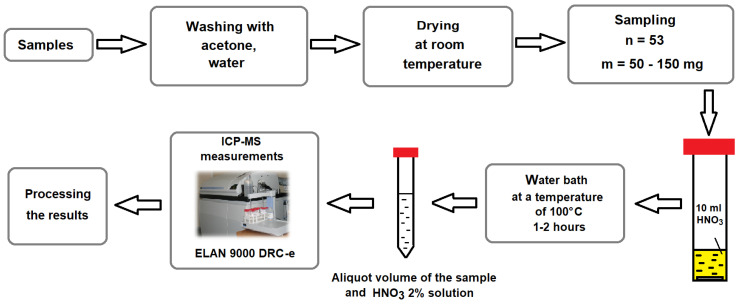
General graphical diagram of this study.

**Figure 2 biology-13-00626-f002:**
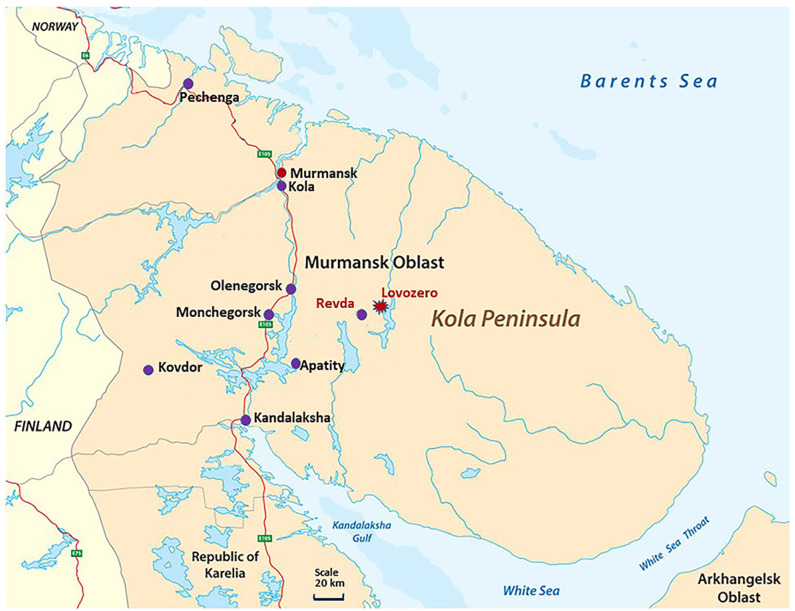
Main sources of environmental pollution in the Kola Peninsula (purple dots), including the Revda settlement with the mining site. The Lovozero village (mark by red) locates in the vicinity of the mining area. Figure modified by the authors based on the map from https://www.worldatlas.com/peninsulas/kola-peninsula.html (accessed on 26 July 2024).

**Figure 3 biology-13-00626-f003:**
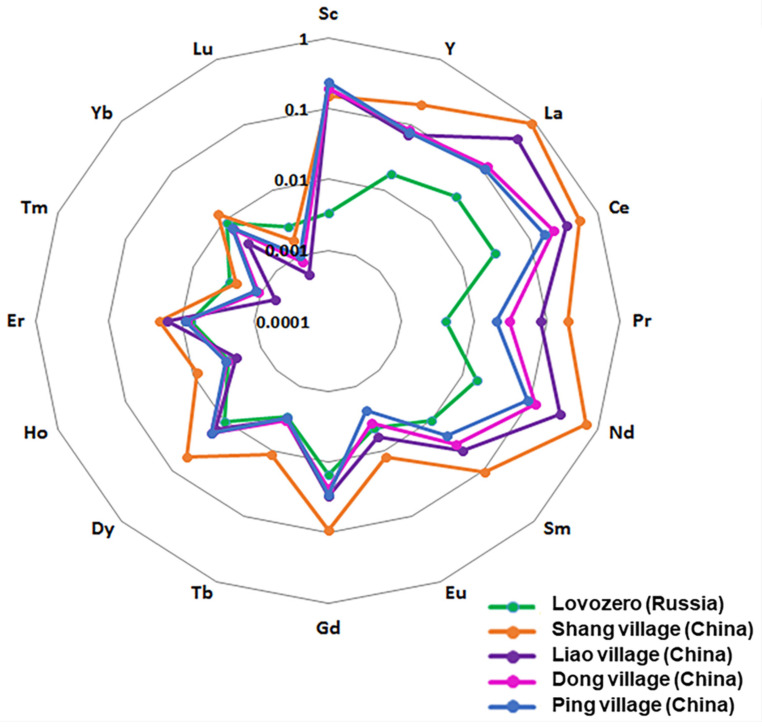
REEs (*G.M.*) in hair samples from different villages in China (M1—Shang village, M2—Liao village, C1—Dong village, C2—Ping village, see text) and from Lovozero village (L) in Russia (Murmansk region), µg/g.

**Figure 4 biology-13-00626-f004:**
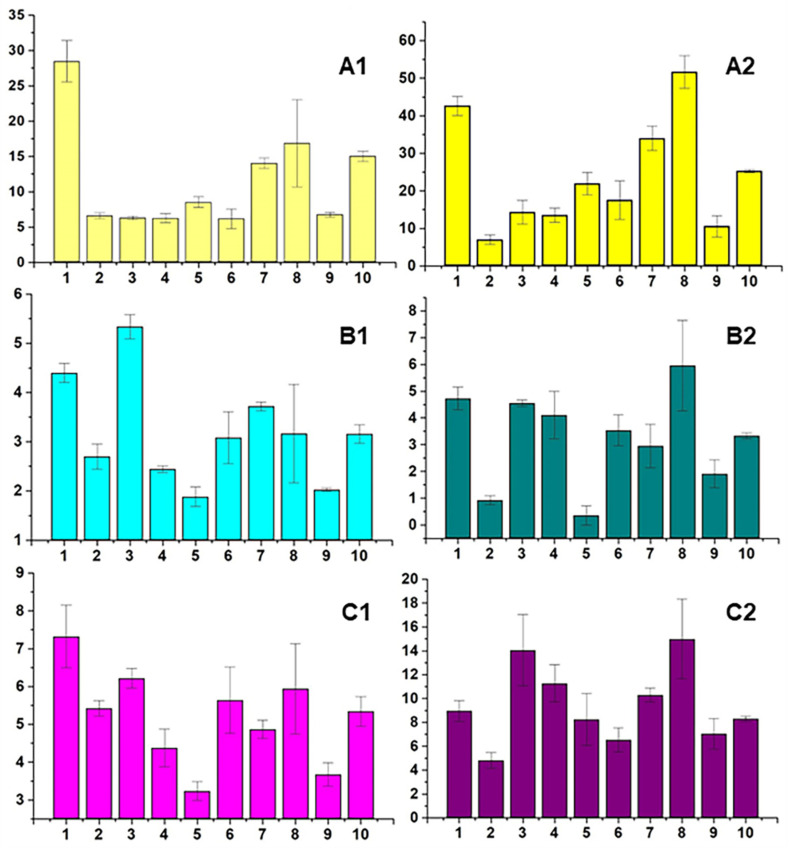
Average annual values (*M ± SE*) of the prevalence of nervous system diseases in children (left column—0–14, right column—15–17 years old) for the period of 2018–2020: (**A1**,**A2**) episodic paroxysmal disorders (G40–G47); (**B1**,**B2**) cerebral palsy and other paralytic syndromes (G80–G83); (**C1**,**C2**) epilepsy and status epilepticus (G40–G41). The abscissa axis compares territories, cities, and districts: 1—Murmansk; 2—Apatity; 3—Kandalaksha; 4—Monchegorsk; 5—Olenegorsk; 6—Kovdorsky; 7—Kolsky; 8—Lovozersky; 9—Pechengsky; 10—the Murmansk region as a whole. The ordinate axis provides values of the prevalence of nervous system diseases (per 1000 corresponding age population).

**Figure 5 biology-13-00626-f005:**
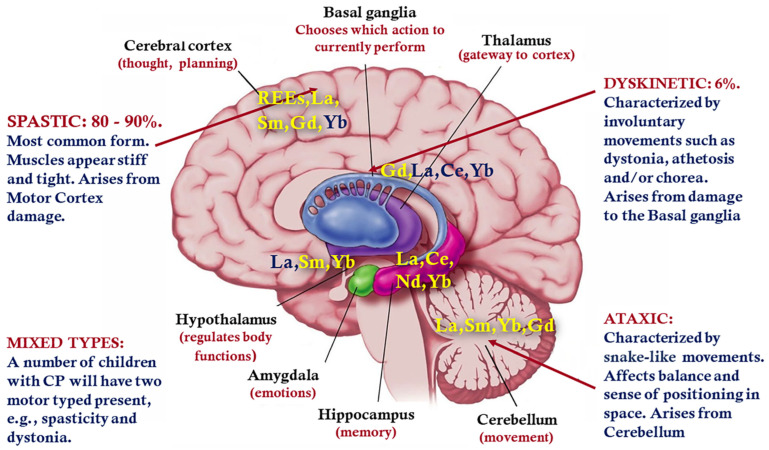
The location of brain regions responsible for motor disorders and the accumulation of certain REEs in these structures. REEs, which hypothetically accumulate in the relevant brain structures, is labelled in royal blue. Figure modified by the authors based on a model of the brain from https://i.pinimg.com/originals/f3/22/b0/f322b02e21b31540ff702e91ad4db722.jpg, accessed on 17 June 2024.

**Table 1 biology-13-00626-t001:** Statistical characteristics of the content of REEs (µg/g) in hair samples from children living in Lovozero village in the vicinity of the mining facility.

*n* = 53	*M*	*SD*	*Min*	*Max*	*Me*	*P25*	*P75*	*C. V.*
Sc	0.0536	0.1238	0.0010	0.5544	0.0010	0.0010	0.0010	230.9
Y	0.0204	0.0152	0.0090	0.0782	0.0157	0.0130	0.0193	74.6
La	0.0336	0.0222	0.0124	0.1288	0.0269	0.0198	0.0362	66.2
Ce	0.0351	0.0219	0.0053	0.1073	0.0297	0.0200	0.0412	62.3
Pr	0.0047	0.0019	0.0020	0.0131	0.0044	0.0038	0.0051	39.3
Nd	0.0161	0.0048	0.0089	0.0383	0.0154	0.0132	0.0174	29.7
Sm	0.0124	0.0069	0.0031	0.0548	0.0116	0.0092	0.0141	55.7
Eu	0.0049	0.0030	0.0017	0.0233	0.0045	0.0036	0.0054	61.9
Gd	0.0214	0.0024	0.0013	0.0630	0.0163	0.0101	0.0277	76.7
Tb	0.0032	0.0009	0.0015	0.0046	0.0032	0.0025	0.0040	27.3
Dy	0.0129	0.0199	0.0055	0.1531	0.0095	0.0080	0.0118	154.2
Ho	0.0031	0.0010	0.0015	0.0058	0.0033	0.0023	0.0038	30.3
Er	0.0077	0.0017	0.0032	0.0118	0.0076	0.0067	0.0091	22.4
Tm	0.0029	0.0009	0.0012	0.0050	0.0027	0.0023	0.0035	31.3
Yb	0.0121	0.0223	0.0037	0.1697	0.0091	0.0072	0.0103	184.0
Lu	0.0028	0.0012	0.0015	0.0089	0.0027	0.0022	0.0033	43.5

**Table 2 biology-13-00626-t002:** Comparisons of some trace element and REE contents (µg/g) in children’s hair samples from Russia (Lovozero village, Murmansk region (1), Zabaikalsky Krai (2)) and Kazakhstan (Pavlodar District, 3) and in adult hair samples from Sweden (4).

Element	1		2		3	4	
	*M ± SD*	*Me (Q25*–*Q75)*	*M ± SD*	*Me (Q25*–*Q75)*	*M ± SE*	*M (SD) ME*	*Range*
Na	642 ± 599	431 (275–783)	728 ± 1198	389(143–940)	223 ± 17	147 (149) 94	17–670
Ca	287 ± 175	220 (186–349)	960 ± 2844	228(100–673)	1403 ± 90	750 (660) 590	113–2890
Sc	0.054 ±0.124	0.001 (0.001–0.001)	0.02 ± 0.01	0.02(0.01–0.02)	0.007 ± 0.0004	0.0014 (0.001) 0.0011	0.0004–0.0045
Cr	4.21 ±13.64	4.43 (1.32–6.5)	5.23 ± 5.69	4.68(0.46–6.11)	0.5 ± 0.07	0.167 (0.118) 0.131	0.046–0.527
Fe	24.5 ±18.6	23.0 (13.6–29.8)	853 ± 872	793(300–1018)	45 ± 3.7	9.6(4.4) 8.4	4.9–23
Co	0.030 ±0.043	0.008 (0.002–0.047)	1.05 ± 1.42	0.66(0.53–0.9)	0.07 ± 0.01	0.013 (0.011) 0.01	0.002–0.063
Zn	120 ±89	106 (77–130)	210 ± 348	152(117–194)	207 ± 8	142 (29) 144	68–198
As	0.61 ±0.70	0.45 (0.001–0.96)	0.42 ± 1.02	0.24(0.16–0.5)	<0.8	0.085 (0.054) 0.067	0.034–0.319
Se	0.71 ± 0.28	0.68 (0.50–0.87)	-	-	0.8 ± 0.03	0.83 (0.28) 0.79	0.48–1.84
Br	-	-	8.82 ± 17.23	3.96(2.07–7.91)	6.5 ± 0.6	37 (33) 26	5.6–221
Rb	0.72 ± 0.79	0.48 (0.27–1.05)	0.79 ± 1.24	0.3(0.2–0.81)	<3	0.093 (0.085) 0.06	0.012–0.482
Sr	1.38 ±1.65	0.59 (0.32–2.06)	4.53 ± 1.23	5(2.5–5)	<15	1.2 (1) 0.97	0.14–5.54
Ag	0.28 ±0.16	0.23 (0.19–0.35)	2.34 ± 17.67	0.11(0.05–0.26)	0.3 ± 0.03	0.231 (0.298) 0.132	0.025–1.96
Sb	0.06 ±0.06	0.044 (0.028–0.07)	0.11 ± 0.15	0.06(0.03–0.12)	0.07 ± 0.01	0.022 (0.017) 0.017	0.007–0.122
Cs	0.014 ±0.022	0.01 (0.005–0.014)	0.03 ± 0.14	0.005(0.001–0.02)	<0.05	0.00067 (0.00046) 0.00051	0.00017–0.0019
Ba	0.58 ±0.88	0.41 (0.21–0.60)	4.07 ± 12.24	2(1.97–2.43)	<10	0.64 (0.49) 0.46	0.16–1.92
Y	0.020 ±0.015	0.016 (0.013–0.019)	-	-	-	-	-
La	0.034 ±0.022	0.027(0.02–0.036)	0.04 ± 0.09	0.02(0.01–0.02)	0.05 ± 0.006	0.035 (0.046) 0.018	0.0046–0.106
Ce	0.035 ±0.022	0.03 (0.02–0.041)	0.39 ± 1.79	0.1(0.05–0.1)	<0.08	0.039 (0.05) 0.019	0.007–0.164
Pr	0.005 ±0.002	0.004 (0.004–0.005)	-	-	-		
Nd	0.016 ±0.005	0.015 (0.013–0.017)	0.25 ± 0.46	0.09(0.09–0.22)	-		
Sm	0.0124 ±0.007	0.012 (0.009–0.014)	0.006 ± 0.02	0.003(0–0.003)	0.02 ± 0.003		
Eu	0.005 ±0.003	0.004 (0.004–0.005)	0.02 ± 0.07	0.004(0.001–0.01)	<0.03		
Gd	0.0214 ±0.016	0.016 (0.010–0.028)	-	-	-		
Tb	0.003 ±0.001	0.003 (0.002–0.004)	0.01 ± 0.02	0.005(0.005–0.02)	0.01 ± 0.0003		
Dy	0.013 ±0.02	0.01 (0.008–0.012)	-	-	-		
Ho	0.003 ±0.001	0.003 (0.002–0.004)	-	-	-		
Er	0.008 ±0.002	0.008 (0.007–0.009)	-	-	-		
Tm	0.003 ± 0.001	0.003 (0.002–0.004)	-	-	-		
Yb	0.012 ±0.022	0.009 (0.007–0.010)	0.007 ± 0.009	0.005(0–0.009)	0.03 ± 0.0005		
Lu	0.003 ±0.001	0.003 (0.002–0.003)	0.003 ± 0.01	0.001(0–0.003)	0.002 ± 0.0001		
Hf	0.004 ±0.005	0.004 (0.001–0.007)	0.05 ± 0.12	0.037(0.001–0.05)	0.02 ± 0.002	0.005 (0.008) 0.0017	0.0004–0.037
Ta	0.006 ±0.015	0.006 (0.001–0.001)	0.02 ± 0.04	0.008(0–0.008	<0.03	0.004 (0.003) 0.0031	<0.002–0.020
Au	0.012 ±0.021	0.012 (0.001–0.017)	0.03 ± 0.14	0.001(0–0.01)	0.11 ± 0.03	0.03 (0.028) 0.017	0.003–0.200
Hg	0.42 ± 0.197	0.42 (0.31–0.5)	-	-	0.4 ± 0.04	0.261 (0.145) 0.249	0.053–0.927
Th	<0.001 <0.001	<0.001	0.23 ± 0.29	0.14(0.05–0.27)	0.02 ± 0.001	0.0013 (0.001) 0.001	0.0003–0.0044
U	0.005 ±0.003	0.005 (0.003–0.006)	0.01 ± 0.01	0.01(0.01–0.016)	0.33 ± 0.07	0.057 (0.065) 0.036	0.006–0.436

**Table 3 biology-13-00626-t003:** Comparisons of REE content, ng/g, except where noted (*), in children’s hair samples from Lovozero village (L) in Russia (Murmansk region) and children’s hair samples from different villages in China (M1, M2, C1, and C2—see text).

	L—*G.M.(R)*	L—*M ± SE*	M1—*G.M.(R)*	M2—*G.M.(R)*	C1—*G.M.(R)*	C2—*G.M.(R)*
La *	0.03 (0.01–0.13)	0.033 ± 0.003	0.89 (0.17–6.93)	0.45 (0.14–0.37)	0.12 (0.05–0.30)	0.11 (0.04–0.40)
Ce *	0.03 (0.01–0.11)	0.035 ± 0.003	0.53 (0.20–1.37)	0.34 (0.16–0.82)	0.22 (0.09–0.57)	0.16 (0.07–0.44)
Pr *	0.004 (0.00–0.01)	0.005 ± 0.000	0.19 (0.04–1.55)	0.08 (0.03–0.26)	0.03 (0.01–0.09)	0.02 (0.01–0.09)
Nd *	0.016 (0.01–0.04)	0.016 ± 0.001	0.67 (0.13–5.27)	0.28 (0.09–0.95)	0.12 (0.04–0.32)	0.09 (0.04–0.32)
Sm *	0.01 (0.00–0.05)	0.012 ± 0.001	0.11 (0.02–0.91)	0.04 (0.03–0.12)	0.03 (0.01–0.06)	0.02 (0.01–0.07)
Eu	4.4 (1.70–23.33)	4.9 ± 0.04	12.1 (2.9–95.0)	6.0 (2.6–21.2)	3.7 (1.8–8.7)	2.4 (1.9–6.6)
Gd	15.1 (1.29–63.22)	21.4 ± 0.2	90.5 (19.4–645.6)	30.6 (12.2–119.0)	23.7 (8.3–59.3)	29.5 (9.8–64.6)
Tb	3.0 (1.51–4.65)	3.17 ± 0.012	11.1(2.6–73.8)	3.3 (1.4–13.2)	3.4 (1.3–8.3)	3.1 (1.6–9.7)
Dy	10.3 (5.51–153.10)	12.9 ± 0.27	53.5 (13.6–338.6)	15.1(6.4–63.0)	18.2 (7.5–44.1)	17.8 (9.8–54.5)
Ho	3.0 (1.53–5.83)	3.14 ± 0.013	8.5 (2.2–55.63)	2.3 (1.0–10.7)	3.1 (1.2–7.3)	3.2 (1.7–10.4)
Er	7.5 (3.23–11.79)	7.74 ± 0.024	20.7 (5.7–134.3)	15.9 (2.7–27.3)	8.5 (3.8–20.1)	8.8 (5.2–28.9)
Tm	2.8 (1.24–4.96)	2.9 ± 0.012	2.3 (0.6–16.2)	0.6 (0.3–3.1)	1.1 (0.4–2.5)	1.2(0.6–4.1)
Yb	9.1 (3.68–169.73)	12.1 ± 0.307	13.5 (4.4–90.5)	3.6 (1.6–18.3)	6.6 (3.6–15.0)	7.1 (3.9–22.3)
Lu	2.7 (1.47–8.93)	2.8 ± 0.017	1.7 (0.4–13.3)	0.5 (0.2–2.6)	0.8 (0.4–2.0)	1.0 (0.4–3.3)
Y	17.5 (0.009–0.0078) *	20.0 ± 2.0	203.5 (0.06–1.27) *	71.2 (0.03–0.30) *	82.3 (0.03–0.19) *	76.5 (0.04–0.29) *
Sc	3.4 (1.0–554.0)	53.6 ± 17.0	154	195.5	194.7	243.1

* Unit µg/g; other—ng/g; *G.M*.: geometric mean; *R*: minimum and maximum values.

**Table 4 biology-13-00626-t004:** Comparisons of the REE content (μg/g) in hair samples of children from Lovozero village (Murmansk region, Russia) and in hair samples of 95 children from four villages around the mining area of the Maoniuping rare earth deposit in Mianning City, Sichuan Province, China.

	Lovozero Village, Russia			4 Village, China		
	*M*	*P25*	*P75*	*M*	*P25*	*P75*
La	0.0269	0.0198	0.0362	0.3907	0.2399	1.1505
Ce	0.0297	0.0200	0.0412	0.5958	0.3811	1.6591
Pr	0.0044	0.0038	0.0051	0.0489	0.0337	0.1271
Nd	0.0154	0.0132	0.0174	0.1646	0.1098	0.3574
Sm	0.0116	0.0092	0.0141	0.0226	0.0150	0.0364
Eu	0.0045	0.0036	0.0054	0.0060	0.0042	0.0097
Gd	0.0163	0.0101	0.0277	0.0184	0.0118	0.0296
Tb	0.0032	0.0025	0.0040	0.0020	0.0009	0.0030
Dy	0.0095	0.0080	0.0118	0.0112	0.0066	0.0151
Ho	0.0033	0.0023	0.0038	0.0020	0.0012	0.0027
Er	0.0076	0.0067	0.0091	0.0051	0.0028	0.0071
Tm	0.0027	0.0023	0.0035	0.0009	0.0005	0.0015
Yb	0.0091	0.0072	0.0103	0.0049	0.0030	0.0064
Lu	0.0027	0.0022	0.0033	0.0006	0.0003	0.0010
Y	0.0157	0.0130	0.0193	0.0497	0.0312	0.0672

## Data Availability

The original contributions presented in this study are included in this article; further inquiries can be directed to the corresponding author/s.

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
