# Peer review of "Rare Earth Element Content in Hair Samples of Children Living in the Vicinity of the Kola Peninsula Mining Site and Nervous System Diseases"

_biology, 2024, doi:10.3390/biology13080626_

Round 1
Reviewer 1 Report
Comments and Suggestions for Authors
The aim of this study is to assess the content of rare earth elements (REE) in hair samples of children living in Lovozero village, near an REE mining site, and to explore the possible effects of REEs on the prevalence of nervous system diseases in the Lovozersky district (Murmansk region, Kola Peninsula). This paper has practical value and is recommended for revision in the following aspects:
1. In the Experimental Procedure section, a complete flowchart could be used to illustrate the process.
2. For the statistical characteristics of REE content, F-tests and N-tests could be employed to highlight differences between methods.
3. In the conclusion, where the need for further research is mentioned, the specific types of research needed could be listed.
4. The simple summary and abstract could be combined, but the abstract should also address the research challenges and motivations, such as a comparative analysis of nervous system and paroxysmal disorders.
5. The clarity of the images needs to be improved.
Minor editing of English language required.
Author Response
Deeply respected Reviewers,
Please accept our sincere gratitude for your contributions to improve the article. Thank you very much!
Reviewer 1
#Comment 1:
1. In the Experimental Procedure section, a complete flowchart could be used to illustrate the process.
Response: Thank you for your helpful advice, a flowchart is included in the text (Fig.1).
#Comment 2:
2. For the statistical characteristics of REE content, F-tests and N-tests could be employed to highlight differences between methods.
Response: Thank you for the interesting proposal. However, in our case we do not detect significance of differences between methods and samples. We demonstrate statistical data on REE content in the study sample (53 persons). It is important for us to show absolute values of measurements. In addition, we compare our data with literature data, where REE content in hair samples of children was dependent on geochemical specificity their regions. And most importantly - we would like show that living of children near REE mines is associated with high incidence of diseases of the nervous system.
#Comment 3: In the conclusion, where the need for further research is mentioned, the specific types of research needed could be listed.
Response: Thanks for the positive advice: a paragraph has been added to the “Conclusion” section that lists the desired studies, marked by red.
#Comment 4: The simple summary and abstract could be combined, but the abstract should also address the research challenges and motivations, such as a comparative analysis of nervous system and paroxysmal disorders.
Response: Thank you for your constructive enquiry. We have only followed the guidelines of journal Biology, according to which we prepared the "Simple Summary" and the "Abstract". In the abstract, the aim of the study is stated. The relevant parts of the journal guidelines are given below.
“Manuscript Preparation
Front Matter
- Simple Summary: It is vitally important that scientists are able to describe their work simply and concisely to the public, especially in an open access online journal. A simple summary consists of no more than 200 words in one paragraph and contains a clear statement regarding the problem addressed, the aims and objectives, pertinent results, conclusions from the study and how they will be valuable to society. This should be written for a lay audience, i.e., no technical terms without explanations. No references should be cited and no abbreviations should be used. Submissions without a simple summary will be returned directly. An example of a simple summary can be found at the following link: https://www.mdpi.com/2079-7737/10/2/139.
- Abstract: The abstract should be a total of about 200 words maximum. The abstract should be a single paragraph and should follow the style of structured abstracts, but without headings: 1) Background: Place the question addressed in a broad context and highlight the purpose of the study; 2) Methods: Describe briefly the main methods or treatments applied. Include any relevant preregistration numbers, and species and strains of any animals used; 3) Results: Summarize the article's main findings; and 4) Conclusion: Indicate the main conclusions or interpretations. The abstract should be an objective representation of the article: it must not contain results which are not presented and substantiated in the main text and should not exaggerate the main conclusions.”
#Comment 5: The clarity of the images needs to be improved.
Response: We completely agree with you and we have significantly improved the clarity of the images (see in the text)
Thank you very much!

Reviewer 2 Report
Comments and Suggestions for Authors
This work requires major revision before publication
1. Justification for need of this research must be delivered concisely in the introduction part. Currently, introduction seems to be more elaborate without clear justification.
2. Why only 53 persons chosen for this interrogations?
3. Is this quantification tactic was compared with other tactic? If yes, clarify for the readers.
4. Too many data in the tables, without any proper discussion.
5. Advantages of this research must be stated before conclusion section.
6. Reference section must be updated with latest relevant literature.
Comments on the Quality of English LanguageMinor English editing is required
Author Response
Deeply respected Reviewers,
Please accept our sincere gratitude for your contributions to improve the article. Thank you very much!
Reviewer 2
#Comment 1: Justification for need of this research must be delivered concisely in the introduction part. Currently, introduction seems to be more elaborate without clear justification.
Response: Heartfelt appreciation for the very important advice. We have shortened and transformed the text of the justification, leaving only what is relevant to our article (inserts are marked by red, those to be excluded are marked by blue).
#Comment 2: Why only 53 persons chosen for this interrogations?
Response: Rural settlements on the Kola Peninsula are small, as is the number of children in them. In particular, there is only one school in the Lovozero village, and the 53 persons are a number of recruited children, whose parents had signed consent for their children to participate in the research.
#Comment 3: Is this quantification tactic was compared with other tactic? If yes, clarify for the readers.
Response: We thank you for this excellent question. It allowed us to further interpret the data contained in Table 2. In our study we applied the Inductively coupled plasma mass spectrometry (ICP-MS) method for elemental analysis, which is widely used in I.V. Tananaev Institute of Chemistry and Technology of Rare Elements and Mineral Raw Materials of Federal Scientific Centre "Kola Science Centre", Laboratory of Chemical and Optical Methods of Analysis. However, our colleagues at National Research Tomsk Polytechnic University, Nuclear geochemical laboratory of the International Innovative Research and Education Centre "Uranium Geology" - use the method of multi-element instrumental neutron activation analysis (INAA).
Table 2 shows data on the content of elements in hair samples from children obtained by these two methods. Lovozero (1), Sweden (4) - ICP-MS; Zabaykalsky region (2) [43]; Kazakhstan (Pavlodar District) [44]- INAA. The comparability of these methods can be seen, for example, when comparing the chromium (Сr) content of children in Lovozero and in Zabaykalsky region: 4.43 (1.32-6.5) and 4.68(0.46-6.11) µg/g, respectively. This is 8.9 and 9.4 times higher than in hair samples of children from Kazakhstan and 26.5 and 28.0 times higher, respectively, than in hair samples of Swedish residents. The increased Cr content in hair samples of children from Lovozero and Zabaikalian are caused by chromitoniferous provinces and deposits of chromium ores in the Karelian-Kola Province and in the Zabaikalian chromitoniferous province. Therefore, the data of Table 2 indirectly testify to the comparability of the two methods of analysis.
However, in our paper we do not aim to prove the comparability of these methods. We demonstrate in Table 2 how natural and anthropogenic features of the territories influence the content of REEs and, thus, show that the elemental composition of hair of children from Lovozero is caused by REEs intake from the environment. And the source of these REEs is Lovozero Mining and Processing Plant. Nevertheless, in view of your highlighted interest of readers in the comparability of different methods of element analysis, we have included this fragment in the content of the “Results” section (marked by red). Thank you very much.
#Comment 4: Too many data in the tables, without any proper discussion.
Response: Thank you very much for your valuable comment. We have substantially revised the Results section. New fragments are highlighted in red, what should be deleted is highlighted in blue.
#Comment 5: Advantages of this research must be stated before conclusion section.
Response: Thank you for the very valuable advice. The ‘Advantages of this research’ section has been included in the text before the ‘Conclusion’ section and is labelled in red .
#Comment 6: Reference section must be updated with latest relevant literature.
Response: Thanks for the helpful advice. We have included important reference for our article in the citation list:
Zielińska-Dawidziak M, Czlapka-Matyasik M, Wojciechowska Z, Proch J, Kowalski R, Niedzielski P. Rare Earth Elements Accumulation in the Hair of Malagasy Children and Adolescents in Relation to Their Age and Nutritional Status. Int J Environ Res Public Health. 2022 Jan 1;19(1):455. doi: 10.3390/ijerph19010455. PMID: 35010715; PMCID: PMC8744718.
As for the rest of the ‘latest relevant literature’ we cite in our study only those works that are directly relevant to our research. And, despite the fact that our article is not a review, the citation list includes 130 literature sources, among which 31 sources reflect the most recent studies over 5 years. We also cite unique papers from the 70-80s, in which jewelry studies on individual axons were performed and the mechanisms of lanthanides' effects on the conduction of nerve impulses were shown. To this day, these works have no equal.
The following is a list of papers from the last five years in the cited literature.
- Wang, W.; Yang Y.; Wang D.; Huang L. Toxic Effects of Rare Earth Elements on Human Health: A Review. 2024, 12, 5, p.317. doi: 10.3390/toxics12050317. PMID: 38787096.
- Brouziotis, A.A.; Giarra, A.; Libralato, G.; Pagano, G.; Guida, M.; Trifuoggi, M. Toxicity of rare earth elements: An overview on human health impact. Environ. Sci. 2022. 10, p.948041. doi: 10.3389/fenvs.2022.948041.
- Balaram V. Rare earth elements: A review of applications, occurrence, exploration, analysis, recycling, and environmental impact. Geoscience Frontiers. 2019, 101285e1303. https://doi.org/10.1016/j.gsf.2018.12.005).
- Li Z, Liang T, Li K, Wang P. Exposure of children to light rare earth elements through ingestion of various size fractions of road dust in REEs mining areas. Sci Total Environ. 2020, 15, pp.743:140432. doi: 10.1016/j.scitotenv.2020.140432. Epub 2020 Jun 23. PMID: 32659548.
- Abdelnour, S.A.; Abd El-Hack, M.E.; Khafaga, A.F.; Noreldin, A.E.; Arif, M.; Chaudhry M.T., Losacco, C.; Abdeen, A.; Abdel-Daim, M.M. Impacts of rare earth elements on animal health and production: Highlights of cerium and lanthanum. Sci Total Environ. 2019, 1, 672, pp.1021-1032. doi: 10.1016/j.scitotenv.2019.02.270. PMID: 30999219
- Gaman, L.; Radoi, M.P.; Delia, C.E.; Luzardo, O.P.; Zumbado, M.; Rodríguez-Hernández, Á.; Stoian, I.; Gilca, M.; Boada, L.D.; Henríquez-Hernández, L.A. Concentration of heavy metals and rare earth elements in patients with brain tumours: Analysis in tumour tissue, non-tumour tissue, and blood. J. Environ. Health Res. 2021, 31, pp.741–754. DOI: 10.1080/09603123.2019.1685079
- Zheng L, Zhang J, Yu S, Ding Z, Song H, Wang Y, Li Y. Lanthanum Chloride Causes Neurotoxicity in Rats by Upregulating miR-124 Expression and Targeting PIK3CA to Regulate the PI3K/Akt Signaling Pathway. Biomed Res Int. 2020, 5, p.5205142. doi: 10.1155/2020/5205142. PMID: 32461997; PMCID: PMC7222569.
- Wei J., Wang C., Yin S., Pi X., Jin L., Li Z. et al.) Concentrations of rare earth elements in maternal serum during pregnancy and risk for fetal neural tube defects. Environ Int. 2020, 137, p.105542. https://doi.org/10.1016/j.envint.2020.105542.
- Li Z, Liang T, Li K, Wang P. Exposure of children to light rare earth elements through ingestion of various size fractions of road dust in REEs mining areas. Sci Total Environ. 2020, 15, 743, 140432. doi: 10.1016/j.scitotenv.2020.140432. PMID: 32659548.
- Y.,; Wang, D.; Huang Yi, N.; Sheng, P.; Yang, B. Relationship between Rare Earth Elements, Lead and Intelligence of Children Aged 6 to 16 years: A Bayesian Structural Equation Modelling Method. Int. Arch. Nurs. Health. Care. 2019, 5, p. 123. doi.org/10.23937/2469-5823/1510123.
- Cao, Z.; Yang, M.; Gong, H.; Feng, X.; Hu, L., Li; R., Xu, S.; Wang, Y.; Xiao, H.; Zhou, A. Association between prenatal exposure to rare earth elements and the neurodevelopment of children at 24-months of age: A prospective cohort study. Environ Pollut. 2024, 15, 343, p.123201. doi: 10.1016/j.envpol.2023.123201. PMID: 38135135.
- Mazukhina, S.; Krasavtseva, E.; Makarov, D.; Maksimova, V. Thermodynamic Modeling of Hypergene Processes in Loparite Ore Concentration Tailings. 2021, 11, 996. https://doi.org/10.3390/min11090996
- Krasavtseva, E.A.; Maksimova, V.V.; Gorbacheva, T.T.; Makarov, D.V.; Alfertyev, N.L. Evaluation of soils and plants chemical pollution within the area affected by storages of loparite ore processing waste. Mine Surv. Subsurf. Use 2021, 112, 52–58. (In Russian).
- Krasavtseva, E.; Sandimirov, S.; Elizarova, I.; Makarov, D. Assessment of Trace and Rare Earth Elements Pollution in Water Bodies in the Area of Rare Metal Enterprise Influence: A Case Study-Kola Subarctic. Water 2022, 14,3406. https://doi.org/10.3390/w14213406.
- Shin, S.H.; Kim, H-O.; Rim, K.T. Worker Safety in the Rare Earth Elements Recycling Process From the Review of Toxicity and Issues. Safety and Health at Work. 2019, 10, 4, pp. 409-419. https://doi.org/10.1016/j.shaw.2019.08.005.
- Edahbi, M.; Plante, B.; Benzaazoua, M. Environmental challenges and identification of the knowledge gaps associated with REE mine wastes management. J. Clean. Prod. 2019, 212, pp. 1232–1241. https://doi.org/10.1016/j.jclepro.2018.11.228
- Seregina, I.F.; Osipov, K.; Bolshov, M.A.; Filatova, D.G.; Lanskaya, S.Yu. Matrix Interference in the Determination of Elements in Biological Samples by Inductively Coupled Plasma Mass Spectrometry and Methods for Its Elimination. Analyt. Chem. 2019, 74, pp. 182–191. https://doi.org/10. 1134/S1061934819020114.
- Mikhailova, L.A.; Baranovskaya, N.V.; Bondarevich, E.A.; Vitkovsky, Y.A.; Zhornyak, L.V.; Epova, E.S.; Eryomin, O.V.; Nimaeva, B.V.; Ageeva, E.V. Determination of elemental homeostasis of the child population of Zabaikalsky Krai by the method of multi-element instrumental neutron activation analysis. Gigiena i Sanitariya. 2023, 102, 2, pp. 123-131 (in Russian).
- Baranovskaya N.V., Mazukhina S.I., Panichev A.M., VakhЕ.А., Tarasenko I.A., Seryodkin I.V., Ilenok S.S., Ivanov V.V., Ageeva E.V., Makarevich R.A., Strepetov D.A., Vetoshkina A.V. Features of chemical elements migration in natural waters and their deposition in the form of neocrystallisations in living organisms (physico-chemical modeling with animal testing). Bulletin of the Tomsk Polytechnic University. Geo Assets Engineering, 2024, vol. 335, no. 2, pp. 187–201. DOI:10.18799/24131830/2024/2/4459
- Heuser, J.E. The Structural Basis of Long-Term Potentiation in Hippocampal Synapses, Revealed by Electron Microscopy Imaging of Lanthanum-Induced Synaptic Vesicle Recycling. Cell Neurosci. 2022, 16, pp. 920360. doi: 10.3389/fncel.2022.920360. PMID: 35978856; PMCID: PMC9376242.
- Song, Z.; Mao, H.; Liu, J.; Sun, W.; Wu, S.; Lu, X.; Jin, C.; Yang, J. Lanthanum Chloride Induces Axon Abnormality Through LKB1-MARK2 and LKB1-STK25-GM130 Signaling Pathways. Mol. Neurobiol. 2023, 43, 3, pp. 1181-1196. doi: 10.1007/s10571-022-01237-0.
- Delorme, C.; Giron, C.; Bendetowicz, D.; Méneret, A.; Mariani, L.L.; Roze, E. Current challenges in the pathophysiology, diagnosis, and treatment of paroxysmal movement disorders. Expert Rev Neurother. 2021, 21, 1, pp. 81-97. doi: 10.1080/14737175.2021.1840978. PMID: 33089715.
- Zhang, X.J.; Xu, Z.Y.; Wu, Y.C.; Tan, E.K. Paroxysmal movement disorders: Recent advances and proposal of a classification system. Parkinsonism Relat. Disord. 2019, 59, 131-139. doi: 10.1016/j.parkreldis.2019.02.021. PMID: 30902529.
- Garone, G.; Capuano, A.; Travaglini, L.; Graziola, F.; Stregapede, F.; Zanni, G.; Vigevano, F.; Bertini, E.; Nicita, F. Clinical and Genetic Overview of Paroxysmal Movement Disorders and Episodic Ataxias. J. Mol. Sci. 2020, 21, 10, pp. 3603. doi: 10.3390/ijms21103603. PMID: 32443735; PMCID: PMC7279391.
- Patel, D.R.; Neelakantan, M.; Pandher, K.; Merrick, J. Cerebral palsy in children: a clinical overview. Pediatr. 2020, 9, (Suppl 1), pp. S125-S135. doi: 10.21037/tp.2020.01.01. PMID: 32206590; PMCID: PMC7082248.
- Goyal, R.; Spencer, K.A.; Borodinsky, L.N. From Neural Tube Formation Through the Differentiation of Spinal Cord Neurons: Ion Channels in Action During Neural Development. Mol. Neurosci. 2020, 13, p. 62. doi: 10.3389/fnmol.2020.00062. PMID: 32390800; PMCID: PMC7193536.
- Liu, Y.; Wu, M.; Zhang, L.; Bi, J.; Song, L.; Wang, L.; Liu, B.; Zhou, A.; Cao, Z.; Xiong, C.; Yang, S.; Xu, S.; Xia, W.; Li, Y.; Wang, Y. Prenatal exposure of rare earth elements cerium and ytterbium and neonatal thyroid stimulating hormone levels: Findings from a birth cohort study. Int. 2019, 133 (Pt B), pp. 105222. doi: 10.1016/j.envint.2019.105222. PMID: 31655275.
- Liu, L.; Wang, L.; Ni, W.; Pan Y.; Chen, Y.; Xie, Q.; Liu, Y.; Ren, A. Rare earth elements in umbilical cord and risk for orofacial clefts. Ecotoxicol Environ Saf. 2021, 207, 111284. doi: 10.1016/j.ecoenv.2020.111284. PMID: 32942100.
- Wang, J.; Wu, T.; Ma, L. et al. Action of Akt Pathway on La-Induced Hippocampal Neuron Apoptosis of Rats in the Growth Stage. Res. 2020, 38, pp. 434-446. https://doi.org/10.1007/s12640-020-00206-z.
- Sun, W., Yang, J., Hong, Y. et al. Lanthanum Chloride Impairs Learning and Memory and Induces Dendritic Spine Abnormality by Down-Regulating Rac1/PAK Signaling Pathway in Hippocampus of Offspring Rats. Cell Mol Neurobiol. 2020, 40, 459–475 https://doi.org/10.1007/s10571-019-00748-7
- Engel, T.; Brennan, G.P.; Soreq, H. Editorial: The molecular mechanisms of epilepsy and potential therapeutics. Mol. Neurosci. 2022, 15, pp. 1064121. doi: 10.3389/fnmol.2022.1064121.
Very thank you for your contribution to improve our article.

Reviewer 3 Report
Comments and Suggestions for Authors
Dear authors and editors. I highly appreciate the manuscript of the scientific article "Rare Earth Element Content in Hair Samples of Children Living in the Vicinity of the Kola Peninsula Mining Site and Nervous Systems Diseases". The article is proposed for publication in the journal Biology (ISSN 2079-7737). The article corresponds to the subject of a scientific journal. The article is devoted to an urgent and interesting topic.
There are several comments.
1. Figure 1. Specify the scale of the map, sign the names of the seas, countries (Norway, Finland).
2. Figure 3. Poor quality of the drawing.
3. I recommend that the authors provide a general graphical diagram of the study in section 2.
4. Lines 670-671 duplicate the dates 676-677.
Additional comments:
1. The study addresses the important issue of the impact of the chemical industry and industrial emissions on human health.
2. The article addresses significant gaps in the study of the effects of REEs on the health of children living in industrial areas, which is an insufficiently researched area.
3. The article is dedicated to a unique region of Russia, the Kola Peninsula, where industrial development has a negative impact. This region is underdeveloped and poorly studied. The authors' research makes a significant contribution to the study of this little-known area.
4. The research methodology is described very well. I have no comments on the research methodology and its reproducibility.
5. The conclusions are fully consistent with the presented evidence and arguments.
6. All references provided in the manuscript are appropriate.
7. As I have pointed out earlier, there are deficiencies in the design of Figure 1 (scale, names of geographical features) and the quality of Figure 3.
In general, the article has a high scientific level and can be published after minor comments have been eliminated.
Author Response
Deeply Honored Reviewer,
We are sincerely grateful to you for your appreciation of our work. We have gratefully accepted all your comments and have endeavored to eliminate the shortcomings:
- In the Figure 1 (Now it is Fig.2) we have included the specify the scale of the map, sign the names of the seas and countries (Norway, Finland);
- The quality of Figure 3 and other Figures was improved within our capabilities;
- A general graphical diagram of the study was inserted in the Section 2
- Repetitions in the text according to Lines 670-671 was deleted.
We express our heartfelt gratitude for your review and wish you all the best!

Round 2
Reviewer 2 Report
Comments and Suggestions for Authors
Author's effort on improving the manuscript is appreciated and can be accepted for publication in its current version.